# Functional Traits of a Rainforest Vascular Epiphyte Community: Trait Covariation and Indications for Host Specificity

**Katrin Wagner** [1], **Wolfgang Wanek** [2] and **Gerhard Zotz** [1,3,*]

1  Institute for Biology and Environmental Sciences, Functional Ecology, Carl von Ossietzky University Oldenburg, Box 2503, D-26111 Oldenburg, Germany; katrin.wagner@uol.de
2  Department of Microbiology and Ecosystem Science, Center of Microbiology and Environmental Systems Science, University of Vienna, A-1090 Vienna, Austria; wolfgang.wanek@univie.ac.at
3  Smithsonian Tropical Research Institute, Balboa, Ancon, Panama City 0843-03092, Panama
*  Correspondence: gerhard.zotz@uol.de; Tel.: +49-(0)441-798-3315; Fax: +49-(0)441-798-3331

**Abstract:** Trait matching between interacting species may foster diversity. Thus, high epiphyte diversity in tropical forests may be partly due to the high diversity of trees and some degree of host specificity. However, possible trait matching between epiphyte and host is basically unexplored. Since the epiphytic habitat poses particular challenges to plants, their trait correlations should differ from terrestrial plants, but to what extent is unclear as epiphytes are underrepresented or missing in the large trait databases. We quantified 28 traits of 99 species of vascular epiphytes in a lowland forest in Panama that were related to plant size, leaf, stem, and root morphology; photosynthetic mode; and nutrient concentrations. We analyzed trait covariation, community weighted means, and functional diversity for assemblages on stems and in crowns of four tree species. We found intriguing differences between epiphytes and terrestrial plants regarding trait covariation in trait relations between plant maximal height, stem specific density, specific root length, and root tissue den-sity, i.e., stem and root economic spectra. Regarding host specificity, we found strong evidence for environmental filtering of epiphyte traits, but only in tree crowns. On stems, community weighted means differed in only one case, whereas > 2/3 of all traits differed in tree crowns. Although we were only partly able to interpret these differences in the light of tree trait differences, these findings mark an important step towards a functional understanding of epiphyte host specificity.

**Keywords:** host preference; leaf carbon isotope ratio; leaf dry matter content; leaf thickness; leaf nitrogen concentration; leaf nitrogen isotope ratio; root tissue density; root nutrient concentration; specific leaf area; specific root length

## 1. Introduction

Vascular epiphytes (i.e., non-parasitic tracheophytes that spend their entire life on other plants without soil contact) attain high α-diversity in many tropical forests [1]. The mechanisms behind local coexistence of such a large number of epiphyte species are poorly understood, although there seems to be a high degree of ecological equivalence among epiphytes so that stochastic processes play an important role in shaping the structure of epiphyte assemblages [2]. However, niche-based processes are also relevant as shown by predictable distribution patterns, such as the separation of species along vertical micro-climatic gradients in forest stands [3–5] and an increase in plants with crassulacean acid metabolism (CAM) from wetter to drier forests [6]. Competition, however, does appar-ently not play an important role as it does in terrestrial systems [1]. Host tree specificity represents another possibility for niche separation [7]. Many studies have aimed at demon-strating distributional biases among tree species [8–10] but their existence and importance have remained elusive. Detecting possible host specificity is hampered by the high species diversity in tropical forests and the associated difficulty to attain adequate sample sizes of tree and epiphyte species in field studies.

Any kind of specificity among species must entail some degree of "trait-matching" [11]—in the case of host tree specificity a matching between tree and epiphyte traits. A large number of species-specific tree traits may affect the general growing conditions for epiphytes and thus cause host specificity, for example, bark water-holding capacity [12,13], bark stability [12], or crown density [14,15], although their (relative) importance is still basically unclear [7]. Host specificity caused by species-specific differences in tree traits may leave an imprint on the average trait values of epiphyte assemblages. For example, the relative proportion of epiphytes with succulent leaves should be higher in tree species with open crowns, or epiphytes with more adhesive diaspores may be relatively more abundant on trees with smooth bark. There are many species-specific tree traits that may influence epiphyte performance [7]. For example, water availability is contingent on bark roughness and texture (bark water holding capacity), branching architecture (funneling of stemflow), and crown density (throughfall interception). Crown density also influences light interception and vapor pressure deficits (VPD) in the canopy—with stronger seasonal variation in deciduous trees [16]. Bark roughness and stability influence the attachment of epiphyte diaspores. Bark stability, branch turnover, and tree longevity may be effective filters for epiphyte species with a slow life cycle [17]. Nutrient availability may be influenced by leaf and bark nutrient concentrations and their aptitude for leaching [18,19]. Finally, leaf and bark chemistry could also be allelopathic or affect substrate acidity. The large number of candidate traits that can influence growth conditions of epiphytes and their potential covariation arguably makes a mechanistic understanding of host biases difficult. However, systematically varying growth conditions on different tree species should leave detectable imprints on average trait attributes of their epiphyte assemblages due to acclimation of individuals, filtering of genotypes within populations, and species sorting. Detecting correlations between tree and epiphyte characteristics at the species level would provide an unequivocal demonstration of epiphyte host tree specificity.

Currently, plant trait research is a very active field [20]. Most studies focus on soil-rooted plants, but there is a recent surge of studies that quantify functional response traits (i.e., traits whose attributes depend on environmental conditions) [21] of vascular epiphytes, addressing changes in functional response traits in relation to elevation [22,23], precipitation [24,25], forest management or forest type [26,27], and height above ground [3,24]. However, with the exception of one study [16], host tree specificity has so far not been addressed with a trait-based approach. Moreover, the application of this approach for vascular epiphytes still lags behind the developments of general plant functional trait research with an almost exclusive focus on leaf traits (but see Schellenberger Costa et al. [22], who also quantified plant height and stem specific density, and Einzmann, Schickenberg, and Zotz [24], who focused on anatomical root traits.).

The present study addresses the question of epiphyte host specificity with a trait-based approach. To this end, we studied 28 epiphyte traits in a lowland forest in Panama. Traits included plant size, morphological leaf, stem and root traits, photosynthetic mode, leaf isotope ratios and chlorophyll concentration, and leaf and root nutrient concentrations (Table 1). For many of them, trait–environment relationships have been established for soil-rooted plants. For example, increasing water availability should correlate with lower leaf thickness [28,29], higher specific leaf area and larger leaves [30–32]. We expect similar relationships for epiphytes over the short microclimatic and resource gradients that occur within forest stands—with the caveat that most of the reported trait-environment relationships were observed over much larger environmental gradients. An overview of these expectations is given in Figure 1.

**Table 1.** Summary of the 28 studied traits. For each of the 27 continuous traits, which are sorted by trait types, an abbreviation, the number of studied species (*n*), minimum, mean, standard deviation (SD), and maximum values are given. A 28th trait (PP, CAM vs. C3 pathway) is not shown. Sample size varies because it was not possible to determine all traits in every species. "Minimum", "Mean", "SD", and "Maximum" refer to respective parameters for the whole assemblage, generally determined with average trait values per species. In the case of plant height and leaf area, the species values ($H_{max}$, $LA_{max}$) are the largest values observed in a given species.

| Trait Type | Abbreviation | Trait | Unit | *n* | Minimum | Mean | SD | Maximum |
|---|---|---|---|---|---|---|---|---|
| whole plant | $H_{max}$ | maximum plant height | cm | 89 | 1 | 46 | 39 | 230 |
| | $L_{th}$ | leaf thickness | mm | 99 | 0.05 | 0.69 | 0.67 | 4.04 |
| | SLA | specific leaf area | $mm^2\,mg^{-1}$ | 99 | 2.8 | 18.4 | 13.5 | 66.9 |
| | LDMC | leaf dry matter content | $g\,g^{-1}$ | 99 | 0.03 | 0.18 | 0.10 | 0.59 |
| leaf morphological | $LA_{max}$ | maximum leaf area | $cm^2$ | 94 | 0.1 | 223.6 | 805.7 | 6534.1 |
| | SD | stomatal density | $mm^{-2}$ | 88 | 12 | 52 | 36 | 193 |
| | SL | stomatal guard cell length | um | 88 | 19 | 38 | 12 | 76 |
| | PSCA | percent stomatal complex area | % | 88 | 1.0 | 3.8 | 1.8 | 8.6 |
| | RD | root average diameter | mm | 92 | 0.3 | 1.2 | 1.0 | 6.0 |
| root morphological | SRL | specific root length | $m\,g^{-1}$ | 92 | 0.5 | 18.2 | 27.2 | 193.3 |
| | RTD | root tissue density | $g\,cm^{-3}$ | 92 | 0.05 | 0.22 | 0.14 | 0.75 |
| stem morphological | SSD | stem specific density | $mg\,mm^{-3}$ | 87 | 0.05 | 0.22 | 0.12 | 0.63 |
| isotope ratios | $\delta^{13}C$ | leaf carbon isotope ratio | ‰ | 95 | −36 | −28 | 6 | −12 |
| | $\delta^{15}N$ | leaf nitrogen isotope ratio | ‰ | 95 | −5 | −2 | 1 | 2 |
| | LChl | relative chlorophyll concentration per leaf area | SPAD unit | 87 | 12 | 41 | 12 | 68 |
| | LCC | leaf carbon concentration | % | 95 | 34 | 46 | 4 | 62 |
| | LNC | leaf nitrogen concentration | % | 95 | 0.6 | 1.2 | 0.5 | 2.7 |
| leaf chemical | LPC | leaf phosphorus concentration | % | 96 | 0.03 | 0.12 | 0.08 | 0.50 |
| | LKC | leaf potassium concentration | % | 96 | 0.23 | 2.16 | 1.75 | 12.21 |
| | LMgC | leaf magnesium concentration | % | 96 | 0.14 | 0.66 | 0.43 | 2.14 |
| | LCaC | leaf calcium concentration | % | 96 | 0.24 | 1.66 | 1.10 | 4.80 |
| | RCC | root carbon concentration | % | 90 | 44 | 51 | 4 | 72 |
| | RNC | root nitrogen concentration | % | 90 | 0.19 | 0.87 | 0.55 | 3.53 |
| | RPCs | root phosphorus concentration | % | 89 | 0.01 | 0.07 | 0.04 | 0.18 |
| root chemical | RKC | root potassium concentration | % | 89 | 0.09 | 0.99 | 0.87 | 4.34 |
| | RMgC | root magnesium concentration | % | 89 | 0.05 | 0.41 | 0.30 | 1.25 |
| | RCaC | root calcium concentration | % | 89 | 0.04 | 0.87 | 0.83 | 4.45 |

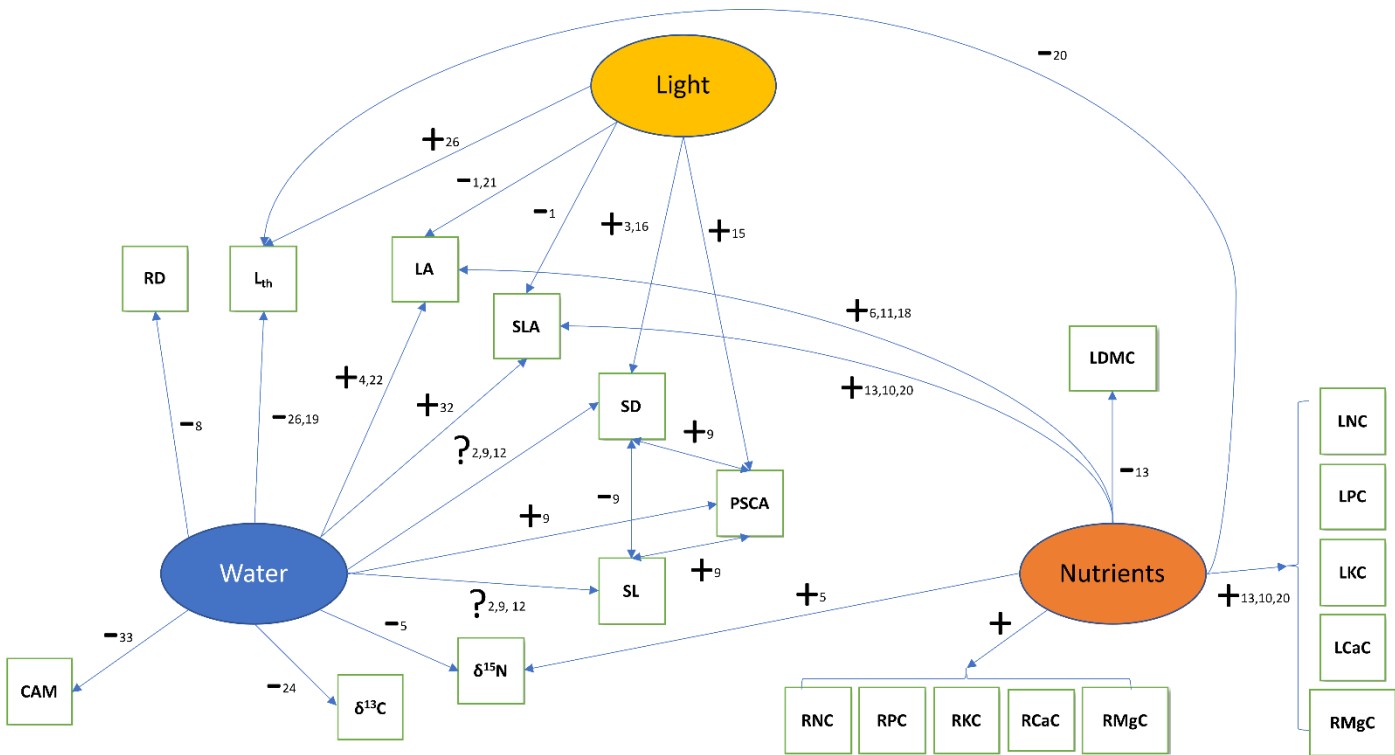

**Figure 1.** Expectations regarding the direction of the trait-environment relationships for epiphytes along (micro-) environmental gradients of water, nutrient, and light availability at the local scale. "+": positive relationship, "−": negative relationship, "?": direction of relationship unclear. Trait abbreviations are explained in Table 1. Numbers refer to publications that established these links and which are listed in the supplement. For a full list of traits, see Table 1.

Members of two important epiphyte families (Orchidaceae and Araceae) often have thick roots with a velamen radicum (a typically multilayered root epidermis composed of dead cells at maturity, [33]), which is thought to allow rapid water uptake and prevent water loss. Therefore, velamen thickness and, thus, root diameter are expected to decrease with improved water availability [34], although there is also conflicting evidence [24]. Expectations regarding stomatal density and size are not straightforward. These traits are negatively correlated with each other and an increase in either of them increases maximal stomatal conductance ($g_{smax}$) which, everything else being equal, should be reduced under water scarcity [35]. However, which of the two parameters drives $g_{smax}$ (here approximated by percent stomatal complex area, PSCA) may depend on the studied system [25,35–37]. Moreover, $g_{smax}$ should also increase in response to light availability [38]. At exposed sites in the canopy, both light intensity and vapor pressure deficit are increased, i.e., water availability is decreased. While in the case of various leaf traits (e.g., specific leaf area, leaf thickness) both environmental factors drive the trait into the same direction from less to more exposed sites (Figure 1), for $g_{smax}$ the two factors lead to opposing trends [39]. Empirical evidence for woody angiosperms indicates that, at the local scale, light intensity has a stronger effect on $g_{smax}$ than VPD. Figure 1 focuses on three abiotic factors: water, light, and nutrients. However, trait matching between tree and epiphyte species may be driven by other tree traits unrelated to these factors. For example, substrate stability (influenced by bark loss rates, wood density, or tree longevity) may filter for fast relative growth rates or small final plant size of epiphytes. Substrate smoothness could be correlated with certain diaspore traits that facilitate attachment to the bark or with root traits related to attachment.

Trait-environment relationships are generally rather weak (e.g., [32]). Along coarse-scale gradients of soil properties and macroclimate there is much unexplained variability,

presumably because these gradients do not adequately capture fine-scale variation in soil conditions and microclimate that actually influence community assembly via species filtering [40]. Moreover, even within any given local plant community there is substantial trait variation, because (1) within relatively homogeneous vegetation stands there are still environmental gradients (e.g., [3]) and (2) for any possible set of abiotic conditions various ecological strategies / trait combinations can be successful [41]. Despite this intra-community variability, community weighted trait means (CWMs) have been repeatedly found to reflect local- or regional-scale environmental gradients (e.g., [30,42,43]). We also use CWMs to infer assemblage differences between tree species.

Apart from average trait values the diversity of functional traits (FD) may also vary among plant communities due to convergence or divergence of traits [44]. How functional diversity responds to gradients of productivity and disturbance is still unclear (e.g., [43,45]). However, in the context of epiphyte assemblages, we would expect trait convergence on tree species that offer adverse conditions to epiphytes (e.g., very open or dense crowns or very smooth bark), while trait divergence should be observed in tree species that offer a large range of abiotic conditions because, e.g., their crown architecture allows for the formation of crotches filled with organic material or gradual light attenuation within the crown.

The strength of the effect of a tree species on epiphyte traits should increase vertically from the stem base to the outer crown [46]. For example, while VPD and light availability in tree crowns may be strongly influenced by foliage density, in the understory these variables are primarily determined by the surrounding vegetation [47]. Conversely, species-specific differences on tree trunks regarding bark structure should be much less pronounced in the crown because of a correlation between bark rugosity and age of trunk and branches. This variation asks for a zone-based sampling approach [48].

One important goal of plant functional trait research is to detect and understand patterns of trait coordination. Wright et al. [32] described a global leaf economics spectrum (LES) in which leaf traits align along a one-dimensional axis from an acquisitive strategy (low longevity but high SLA and mass based nutrient concentrations, photosynthesis, and respiration rates) to a conservative strategy (with high longevity and low values for the other traits). Recently, the focus has shifted to non-foliar traits (e.g., root and stem traits) (e.g., [49–51]) with attempts to integrate several trait types for a holistic understanding of plant ecological strategy [52–54], leading to the postulation of a general "fast-slow" economic spectrum integrating root, stem and leaf traits [53]. This implies a root economics spectrum (RES) analogous to the LES [50], although at least for some plant groups empirical evidence suggests that trait relationships are more complex for underground organs than for leaves [51,55,56]. Figure 2 summarizes reported trait correlations for those traits that were studied in the present work.

Due to the scarcity of trait data of epiphytes it is currently unclear whether the trait correlations in Figure 2, established for terrestrial plants, also apply or whether slope or directionality of trait relationships differ because of some peculiarities of the epiphytic habitat. First, one of the central traits is adult plant height, as it is closely related to competitive ability in a closed vegetation (e.g., [52,57]). However, as epiphytes use other plants as support, their size is uncorrelated to the light environment, which is why Westoby et al. excluded "height-parasites" (climbers and epiphytes) from their theoretical considerations. Second, many epiphytes cling directly to bare bark, i.e., they have no body of substrate in which they could bury their roots. Such roots have to solve the problems of high water-loss through evaporation and intermittent water supply, and we expect substantial differences regarding root trait coordination. Moreover, a vertical or even overhanging substrate will pose very different challenges in terms of attachment.

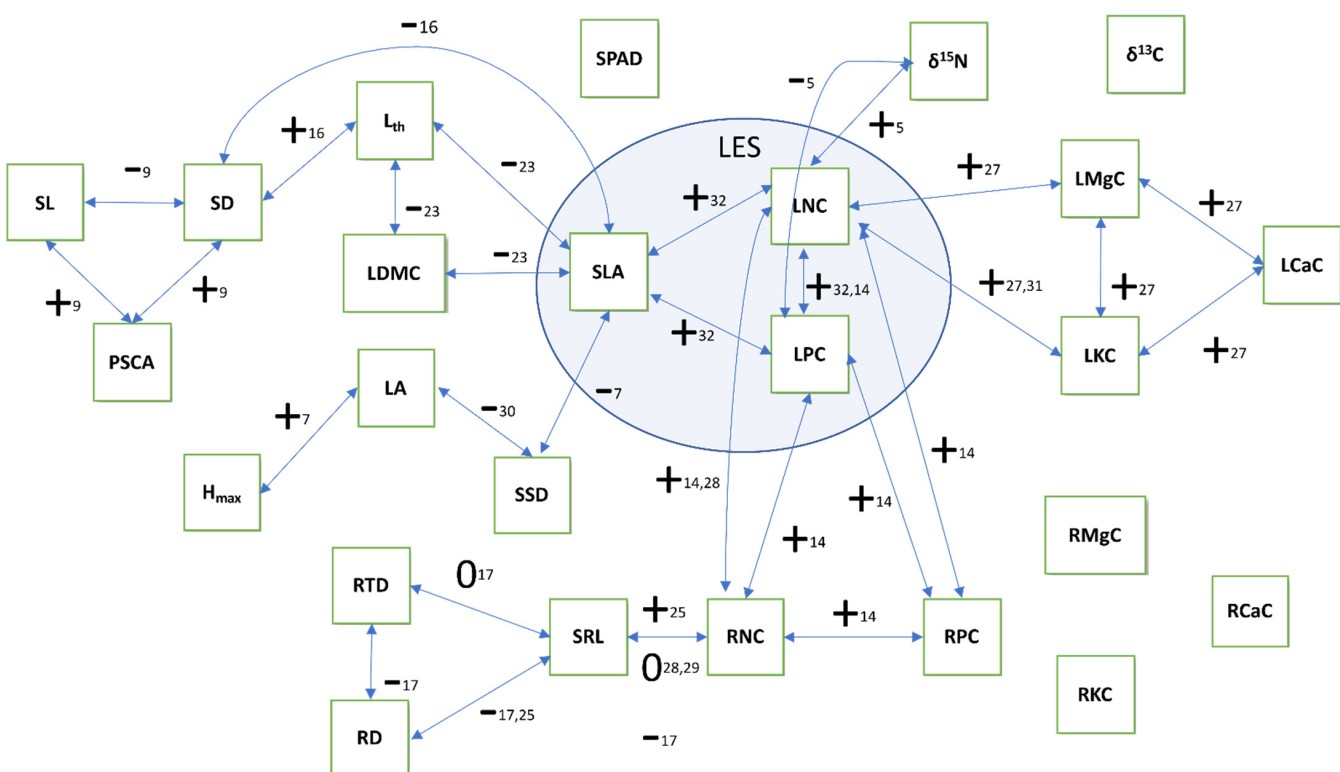

**Figure 2.** Expectations regarding correlations between the studied traits. Trait abbreviations are explained in Table 1. The traits in the inner circle are part of the classical leaf economics spectrum (LES). Arrows indicate reported relationships between traits in the literature. "+": positive relationship, "−": negative relationship, "0": no relationship. Numbers refer to publications that established these links, and which are listed in the supplement. For a full list of traits and abbreviations, see Table 1.

The aims of the present study were threefold. Firstly, to describe broad trait correlations for the epiphyte assemblages at the study site based on a large set of functional traits. This should allow first insights in possible idiosyncrasies regarding epiphyte trait coordination. Secondly, whether tree species exert habitat filtering is assessed by comparing community weighted means of a large range of functional traits among assemblages on different tree species. Third, we compare the functional diversity of epiphyte traits and community-based metrics on different tree species.

## 2. Materials and Methods

### 2.1. Study Site and Focal Tree Species

Sampling for trait measurements and determination of epiphyte abundances on the focal tree species took place in the San Lorenzo forest reserve, a 12,000-ha reserve on the Caribbean slope of Panama (9°17′ N, 79°58′ W, 130 m a.s.l.). Mean annual air temperature and precipitation in this lowland rainforest are 25.2 °C and 3360 mm [58]. A detailed description of the forest is given by Wright et al. [59] and a detailed description of the epiphyte flora can be found in Zotz and Schultz [60]. A 6-ha permanent plot for trees is run by the Center for Tropical Forest Science (CTFS-ForestGeo; [61]), of which 0.9 ha are accessible with a canopy crane. We chose four tree species, all of them tall growing trees (reaching ca. 35–40 m) and among the most abundant species at the study site, which allowed for a large replication: *Aspidosperma spruceanum* Benth. ex Müll.Arg. (Apocynaceae), *Brosimum utile* (Kunth) Oken (Moraceae), *Calophyllum longifolium* Willd. (Clusiaceae) and *Manilkara bidentata* (A.DC.) A.Chev. (Sapotaceae). Per tree species, we sampled 20–21 individuals with a dbh >40 cm, counting the number of epiphytes per species (99 species were found) and Johansson zone (JZ). Johansson zones divide the tree into lower and upper trunk (JZ1 and JZ2) and inner, middle and outer crown (JZ3-JZ5, [46]).

All sampled trees were located within a continuous area of ca. 40 ha. Sampling protocol and composition of the epiphyte assemblages are described in detail in Wagner and Zotz [47]. Several differences among these tree species seem relevant in the context of epiphyte host specificity. Annual dbh increment in *Calophyllum* (0.21 cm year$^{-1}$) is much higher than in the other species (0.12–0.15 cm year$^{-1}$, [47]), whereas the estimated longevity is much lower (253 years; *Aspidosperma*: 468 years, *Manilkara*: 552 years, *Brosimum*: 637 years). Bark rugosity also differs strongly [47]: *Brosimum* bark is smooth, *Aspidosperma* bark is rougher especially in the lower trunk of large trees, with large cracks. Both *Calophyllum* and *Manilkara* possess vertically fissured bark, with large individuals of the latter species having the deepest fissures. Finally, *Brosimum* and *Calophyllum* have more open crowns, while the largest crown extension is found in *Manilkara* [47].

### 2.2. Functional Traits of Epiphytes

We determined 28 functional traits. Depending on the particular trait, data for 87–99 locally occurring epiphyte species could be collected (Table 1). Sampling took place during the rainy season (July-November 2018). We generally aimed at sampling five mature conspecifics, growing in typical microsites. While this goal was even surpassed in some cases, replication was much smaller in rare species. In the extreme case, a species "average" is based on a single observation. To check such values for plausibility, observations were compared to those of other species in the same family. All values were in the expected range. For two species of leafless cacti (*Epiphyllum phyllanthus* (L.) Haw. and *Weberocereus tunilla* (F.A.C. Weber) Britton & Rose) all leaf trait observations were performed on stems [62], except for maximum leaf area.

Plant height was measured in the field as the shortest distance between the point of attachment and the upper or, in pendant species, lower boundary of the main photosynthetic tissue. The maximum plant height ($H_{max}$, Table 1) refers to the maximum observed plant height of a species. Per individual, we collected 1–4 leaves, a section of the stem or rhizome and a representative root sample. To prevent desiccation, samples were transported to the laboratory in closed plastic bags, where plant parts or detached leaves were immediately placed in containers with deionized water for rehydration and root samples were stored at 4 °C. All samples were processed within 24 h.

We removed the petiole from leaves, determined fresh weight, performed the measurements described below and dried samples at 75 °C for >48 h before determining dry weight. Depending on leaf size, a precision scale (Torbal AG500, Scientific Industries Inc., Bohemia, NY, USA), an analytical balance (Torbal AGN 200, Scientific Industries Inc., Bohemia, NY, USA) or a micro-analytical balance (XPE26, Mettler Toledo, Columbus, OH, USA) were used. Leaf dry matter content (LDMC) was calculated as leaf dry weight divided by leaf fresh weight. Photographs of each leaf (EOS 750D, Canon, Tokyo, Japan) allowed area determination with ImageJ [63], which was used to determine specific leaf area (SLA, leaf area/dry weight) for each leaf sample and maximum leaf area ($LA_{max}$) for each species. Leaf thickness ($L_{th}$) represents the average of three measurements with an electronic micrometer (IP 54, Helios-Preisser, Gammertingen, Germany) in the central part of the leaf, avoiding major veins. Leaf chlorophyll concentration per leaf area (LChl) was approximated with an optical meter (SPAD-502, Konica Minolta Sensing, Inc., Sakai, Osaka, Japan), averaging three readings in the central part of the leaf. Stomatal traits were studied with a thin layer of nail polish applied to the central part of the lower and upper sides of leaves. Micrographs of the impressions were taken with a digital microscope camera (DP73) attached to a microscope (BX53) and analyzed with the cellSens Standard software (all Olympus K.K., Tokyo, Japan). We determined stomatal density (SD, averages of three micrographs per impression), length (SL) and width (SW; averages of three stomata per impression). For four amphistomal species, SD was taken to be the sum of the stomatal densities of both leaf sides. Percent stomatal complex area (PSCA) was calculated as PSCA = SD × $\frac{1}{4}$ π (SL × SW) × 100 [64].

Root trait measurements were performed on a representative sample of the entire root system of each plant and not only on fine roots (i.e., diameter < 2 mm), because epiphytes do not have the same clear-cut functional differentiation regarding absorptive fine roots and non-absorptive transport roots as terrestrial plants. Zotz and Winkler [65] showed experimentally that *Phalaenopsis* orchids can take up nutrients over the entire root length and not just at the root tips as in terrestrial plants. Moreover, the roots of many epiphytes have no or little ramifications and in some velamentous species roots invariably exceed 2 mm in diameter. Fresh, cleaned roots were photographed (without overlap), dried at 75 °C for >48 h and weighed. Using WinRHIZO$^{TM}$ (Regent Instruments, Québec, Canada), photographs were used to determine specific root length (SRL, root length/root dry mass), average root diameter (RD) and root tissue density (RTD, root dry mass/root volume).

The volume of fresh stems or stem sections was determined via water displacement [62]. For small samples, we used pycnometers (25 cm$^3$ and 100 cm$^3$, BLAUBRAND$^®$, BRAND GmbH, Wertheim, Germany) to increase accuracy. For species without upright stems, we sampled the rhizome. In the case of orchids with pseudobulbs, we sampled these 'thickened' stems (i.e., thickened stems, [66]). Ideally, sampled stem sections were 10 cm long. When stems or rhizomes were much smaller, several sections were combined for measurements. Afterwards, stems were dried at 75 °C for at least 72 h, dry mass was determined, and stem specific density (SSD) was calculated as stem dry mass divided by the volume of the fresh stem.

For determination of leaf and root nutrient concentrations and foliar isotope ratios, oven-dried samples were ground with a ball mill (small samples: TissueLyser II, QIAGEN N.V., Venlo, Netherlands; larger samples: MM 400, Retsch, Haan, Germany). In a few cases, samples of several individuals had to be combined for chemical analyses. Leaf nitrogen concentration (LNC) and leaf nitrogen ($\delta^{15}$N) and carbon ($\delta^{13}$C) isotope ratios were determined by elemental analyzer-isotope ratio mass spectrometry (Delta PLUS; Thermo Electron, Bremen, Germany). As universal standards, atmospheric air was used for $^{15}$N and the Vienna Pee Dee Belemnite for $^{13}$C. Leaf $\delta^{13}$C was used as a general proxy for leaf water use efficiency (WUE) across C$_3$ and CAM plants because (i) both WUE and $\delta^{13}$C are much higher in CAM species than in C3 species, (ii) $\delta^{13}$C and WUE in CAM species scales positively with the extent of nocturnal CO$_2$ fixation via PEPC, and (iii) $\delta^{13}$C and WUE in C$_3$ species scale also positively with diffusional limitation of photosynthetic CO$_2$ fixation. Based on the $\delta^{13}$C values we also determined one categorical trait, namely photosynthetic pathway (PP), classifying epiphytes in CAM (at least some $\delta^{13}$C values $\geq -21$‰) and C$_3$ species (all $\delta^{13}$C measurements $< -21$‰). Root nitrogen concentration (RNC) was determined with an elemental analyzer with a thermal conductivity detector (Flash EA 1112, Thermo Fisher Scientific, Milano, Italy) according to manufacturer's recommendations and using copper and tungsten oxide as catalysts and acetanilide as standard. To determine leaf and root P, K, Ca and Mg, approximately 5 mg of a sample were digested in 200 μL concentrated HNO$_3$ and 30 μL 30% H$_2$O$_2$ and filled up to 1 mL with distilled water [67]. Leaf (LPC) and root (RPC) phosphorus concentrations were determined colorimetrically in an aliquot of 0.1 mL of the digestion solution [68]. The absorption by the molybdenum-phosphorus complex was measured at 710 nm using a UV-VIS spectrophotometer (Specord 50, Analytik Jena, Jena, Germany). Concentrations of potassium, calcium, and magnesium in leaves and roots (LKC, RKC, LCaC, RCaC, LMgC, RMgC) were determined with a flame atomic absorption spectrometer (240 AA, Agilent technologies, Mulgrave, Australia) with element-specific hollow-cathode lamps and an acetylene flame according to manufacturer's recommendations regarding wavelength and slit settings.

### 2.3. Data Analysis

All statistical analyses were carried out with R version 3.5.2 [69]. Graphs were produced using the R package ggplot2 [70].

To account for the approximate log-normal distribution of most traits, data were log$_{10}$-transformed. Exceptions were LChl, $\delta^{15}$N and $\delta^{13}$C where data were approximately

normally distributed. The proportion of $C_3$ species in an assemblage (PP_C3) was arcsine square root transformed. Species averages were used in all analyses.

To assess bivariate trait relationships, we first run Pearson's correlation analyses visualizing the correlation matrix with the corrplot.mixed function of the corrplot package version 0.84 [71]. To analyze trait coordination in the epiphyte assemblage, we conducted principal component analyses (PCA) with the PCA function of the FactoMineR package version 2.0 [72] scaling variables to unit variance before analysis. PCA graphs were produced with the factoextra package version 1.0.5 [73]. A first PCA was run with a maximum of traits including almost all continuous traits except LChl (i.e., 26 traits). After removing epiphyte species with missing values for any trait, a species-trait-matrix with 72 epiphyte species was obtained. Hymenophyllaceae lack roots and stomata. We thus ran an additional PCA on a dataset without root and stomatal traits, to locate Hymenophyllaceae and some other species with missing observations in the trait space of the remaining 14 traits (81 species).

To calculate community-weighted means (CWMs, e.g., [74]), we defined an epiphyte "community" as the epiphyte assemblage on the trunk or in the crown of an individual tree and divided the epiphyte dataset in a trunk dataset (Johansson zones 1–2) and a crown dataset (Johansson zones JZ3–5). Of the 99 epiphyte species for which we determined trait values, only 76 also occurred on the sampled trees (trunk: 54, crown: 65). Some Hymenophyllaceae were found on the sampled trees but were excluded from analysis because no individual abundance data were available. Likewise, a few epiphyte species occurring on the sampled trees (all rare species accounting together for <1% of the total number of epiphytes in the dataset) were not represented in the traits dataset and therefore also excluded. Only assemblages with at least 10 epiphyte individuals per tree individual (trunk or crown zone) were considered. The resulting number was 20, 21, 17, and 20 crown and 7, 12, 9 and 16 trunk assemblages for *Aspidosperma*, *Brosimum*, *Calophyllum*, and *Manilkara*, respectively. CWMs were calculated with the dbFD function of the FD package [75,76] for all 27 continuous traits plus photosynthetic pathway (PP_C3, a binary trait). In this case, "CWM" refers to the relative abundance of $C_3$ species per assemblage. The dbFD function was also used to calculate the multivariate functional diversity indices "functional dispersion" [75], "functional evenness" and "functional divergence" [77], which describe different facets of functional diversity. Functional dispersion is a measure of the trait spread in multidimensional trait space being "the mean distance in multidimensional traits space of individual species to the centroid of all species." Functional evenness is based on the minimum spanning tree that links all species in the multidimensional trait space and measures the regularity with which the trait space is filled by species, weighted by their abundance. Finally, functional divergence measures the divergence of the abundance distribution in the trait space by using an index that quantifies how species diverge in their distances (weighted by their abundance) from the center of gravity in the trait space.

To assess whether CWMs differed significantly with tree species, we performed PCAs on the CWMs of individual tree assemblages (separately for crown and trunk assemblages) and revised the clustering of trees of the same species. We then tested for significant differences among PC scores of the focal tree species on PC 1 and PC 2.

Besides this multivariate approach, we also tested whether CWMs differed significantly with tree species by performing pairwise comparisons. We visually assessed whether the response variable residuals were approximately normally distributed with Q-Q plots. Whenever Bartlett's test indicated homogeneous variances, we used ANOVA followed by Tukey's tests. Otherwise, Welch's heteroscedastic F test followed by max-t tests for multiple comparisons using the multcomp and sandwich packages [78,79] were used. The same approach was used to test for differences regarding functional diversity indices as a function of host tree species.

## 3. Results

### 3.1. Epiphyte Trait Values and Trait Covariation

Average species values are given for all continuous traits in Table 1 and bivariate correlations between the studied traits in Figure S1. $H_{max}$ showed a pronounced positive correlation with LA (r = 0.79), a weaker positive correlation with LDMC and weak negative correlations with SLA, $L_{th}$ and SRL. Larger leaves tended to be thinner with higher LDMC. SLA had a moderate, negative correlation with its two components, $L_{th}$ and LDMC [62] and, in accordance with the LES, a moderate (LPC) to strong (LNC) positive correlation with leaf nutrients (Figure S3). Moreover, in accordance with a whole-plant "fast-slow" economic spectrum SLA showed a moderate, positive correlation with SRL [53], but no correlation with SSD or SD. SD and SL are inevitably strongly negatively correlated [35, r = −0.74 in our dataset], but PSCA only increased with SD (r = 0.47). Nutrient concentrations of leaves and roots were generally positively correlated among each other and negatively correlated with LCC and RCC (Figure S3). Strong, positive correlations among leaf nutrients were observed for N, P and K (r = 0.47–0.71), Ca and Mg (r = 0.62) and K and Mg (r = 0.39). In contrast, leaf Ca was not correlated with leaf N, P, and K and leaf Mg was uncorrelated to leaf N. Correlations between nutrient concentrations were stronger in roots than in leaves with positive relations among all nutrients, being strongest between root K, P, and Mg (r = 0.80–0.84) and root N and P (r = 0.75). Nutrient concentrations of leaves and roots were also positively correlated, from moderate (K, r = 0.32) to strong (N, r = 0.65). Leaf $\delta^{13}$C increased with $L_{th}$ (r = 0.56) and decreased with leaf N and SLA (r = 0.48 and 0.43). Foliar $\delta^{15}$N increased with LNC (r = 0.54). The only variable correlated with SSD was LDMC (r = 0.24). The strongest bivariate correlation was found for SRL and RD (r = −0.94). SRL was also positively correlated with RTD (r = 0.41), while RD was negatively correlated with RTD (r = −0.69).

A large proportion (65%) of the variation of 26 functional traits was captured by the first four principal components of a PCA (Figure 3A,B, Table S1). The first principal component (PC 1, explained variation 26%) was most closely correlated with mass-based nutrient concentrations—especially in roots. The second principal component (explained variation 18%) was closely correlated to SRL (loading: 0.88) and RD (loading: −0.93), whereas RTD was correlated to both, PC 1 and PC 2 (Figure 3A,B, Table S1). PC 3 (explained variation 12%) was closely correlated to maximal leaf area and plant height. Stomatal density and length were closely correlated to PC 4 (explained variation 9%). Leaf thickness was very well captured by PC 3 and 4, whereas SLA was most closely correlated to PC 2 but almost equally close to PC 1 and 4 and LDMC was best captured by PC 1 and 3. Plant families partly separated in trait space defined by PCs 1–4 (Figure 3C,D). Orchids and especially aroids had low SRL and high RD, due to their velamen radicum, occupying the lower part of the ordination plot spanned by PC 1 and 2. Due to high tissue nutrient concentrations, Piperaceae (all genus *Peperomia*) and Araceae (all genus *Anthurium*) clustered at the right-hand side of the PC 1–2 plot, but separated along PC 2 (i.e., *Peperomia* species had higher SRL).

Running a PCA without stomatal and root traits allowed the inclusion of nine additional species, especially six Hymenophyllaceae. In this PCA, PC 1 (explained variance 28%) was closely correlated to leaf nutrient concentration (e.g., loading of LNC: 0.78, Table S1, Figure S2A), while $L_{th}$ showed a very close correlation with PC 2 (explained variance 22%, loading 0.84). PC 3 was again well described by $LA_{max}$ and $H_{max}$ and PC 4 by SSD (Table S1, Figure S2B). The very thin foliage of filmy ferns led to a clear separation from other families along PC 2, while the high values of leaf P and K in some *Peperomia* species explain again the extreme position along PC 1 (Figure S2C).

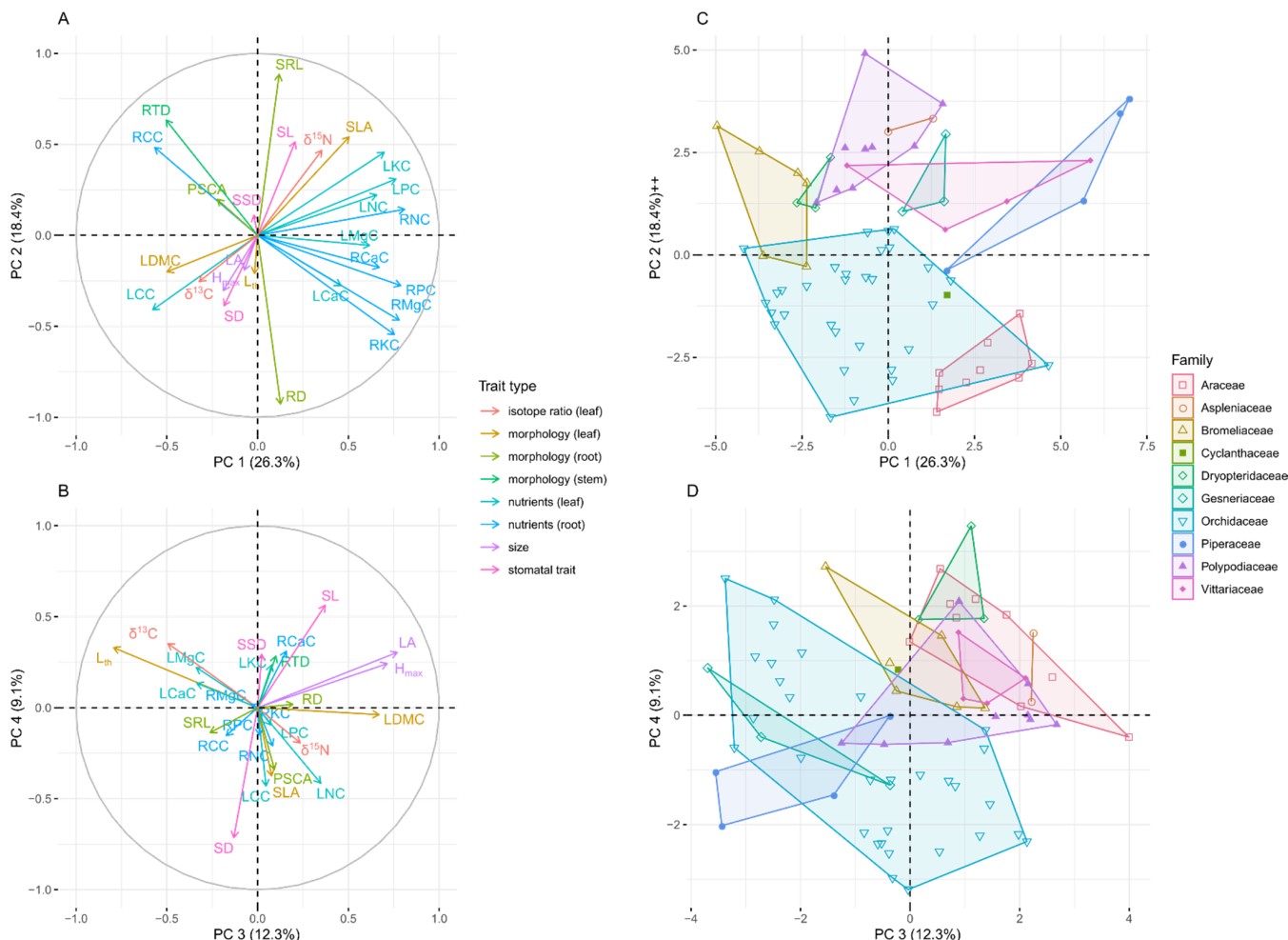

**Figure 3.** Principal component analysis of functional traits of the San Lorenzo epiphyte community. Data are species averages for 26 functional traits of 72 epiphytic species. A and B: Variable correlation plots for the first and second principal component (**A**) and the third and fourth principal component (**B**), respectively. Colors identify trait types. Trait abbreviations are given in Table 1. C and D: Epiphyte species within the ordination space. (**C**) first and second principal component, (**D**) third and fourth principal component. Colors identify plant families, which are enclosed by convex hulls.

### 3.2. Differences among Tree Species

The trait values of crown epiphyte assemblages differed among tree species (Figure 4B, Table S2), the first four PCA axes capturing almost 80% of the variability in the CWMs of 28 traits. PC 1 (explained variability 32%) was most strongly correlated with root nutrient traits (especially root K and Mg, loadings > 0.9, Table S2), as well as RD and RTD (loadings: 0.86 and −0.82), while LKC, SD and SL (loadings: 0.87, −0.93, 0.86) followed by LDMC, SLA and $L_{th}$ (loadings: −0.73, −0.69, and 0.75) were traits that strongly correlated with PC 2 (explained variability 25%). Assemblages on *Brosimum* had significantly lower PC scores along PC 1 than those on the other tree species (ANOVA, $F = 9.94$, $df = 3$, $p < 0.001$, followed by Tukey's Test). Along PC 2, assemblages on *Brosimum* had higher PC scores than those on *Manilkara* (ANOVA, $F = 3.11$, $df = 3$, $p = 0.03$, followed by Tukey's Test).

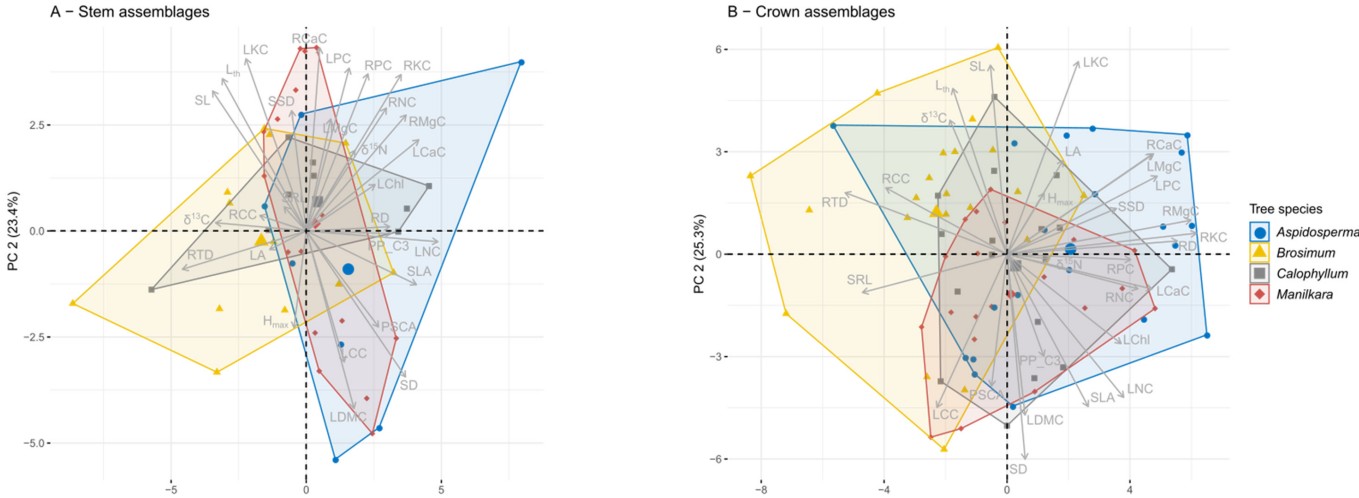

**Figure 4.** Principal component analysis of CWMs of functional traits of the San Lorenzo epiphyte community on tree stems (**A**) and tree crowns (**B**). Analyzed data are based on species averages for 28 functional traits. Trait abbreviations are given in Table 1. Tree species are differentiated by color and enclosed by convex hulls. Group centroids are indicated by larger symbols.

For stem epiphyte assemblages, tree species had no effect on PC scores (Figure 4A, Welch's ANOVA, PC 1: $F = 1.85$, $df = 3$, $p = 0.18$; PC 2: $F = 0.91$, $df = 3$, $p = 0.45$). In line with this finding, the CWMs of the trunk assemblages of the four tree species differed in a single trait: LChl had lower CWMs on *Brosimum* than on *Manilkara* (Table S3, Figure S3). In stark contrast, the CWMs of the crown assemblages differed significantly among tree species for 20 out of 28 traits (Table S3, Figure S4). Significant pairwise differences were most often observed between assemblages on *Brosimum* and *Aspidosperma* (15 traits), followed by *Brosimum* vs. *Calophyllum* and *Brosimum* vs. *Manilkara* (eight traits, Table S2, Figure S3), while pairwise differences between the other three species were rare (*Manilkara-Aspidosperma*, five traits, and *Manilkara-Calophyllum* and *Calophyllum-Aspidosperma*, three traits each). Moreover, CWMs of epiphyte assemblages were most distinct on *Brosimum* (37% of all pairwise comparisons, 31/84 trait comparisons), followed by *Aspidosperma* (27%, 23/84), *Manilkara* (19%, 16/84) and *Calophyllum* (17%, 14/84). *Brosimum* therefore imposed the strongest filtering on their epiphyte assemblage. Regarding leaf morphological traits, assemblages on *Brosimum* had, on average, lower LDMC and higher $L_{th}$ than those on *Aspidosperma* and *Calophyllum*, indicating a higher proportion of species with succulent leaves. Because of higher leaf thickness and lower LDMC, SLA was also lowest on *Brosimum*. Compared to epiphytes on *Aspidosperma*, those on *Brosimum* showed lower CWMs for several nutrients in leaves (N and P) and roots (N, K, and Mg). For leaf N and LChl, this difference was also significant in regard to the other two tree species, while leaf P CWMs of *Calophyllum* and *Manilkara* were not higher than those of *Brosimum*. For the stomatal traits, CWMs of SD were lower, and CWMs of SL were higher on *Brosimum* than on *Manilkara*, but there were no differences for PSCA. Pronounced differences were observed for root traits. CWMs of SRL were highest on *Brosimum* and lowest on *Aspidosperma* while the reverse was true for RD; CWMs of RTD were significantly higher for *Brosimum* compared to *Aspidosperma* and *Manilkara*. Finally, clear differences were observed in the CWMs of $\delta^{13}C$. When all species ($C_3$ and CAM plants) were included, $\delta^{13}C$ CWMs on *Brosimum* were significantly higher than those on *Manilkara*. When only $C_3$ plants were considered ($\delta^{13}C\_2$), *Brosimum* had still the highest values and those of *Aspidosperma* and *Calophyllum* were significantly more negative.

Functional dispersion of crown epiphyte assemblages tended to be larger on *Aspidosperma* (i.e., larger spread in the multidimensional traits space) than on *Brosimum* (p = 0.052, Tukey's Test), while all other pairwise differences were not significant (Figure 5A). Compared to assemblages on the other tree species, functional evenness on *Aspidosperma*

was also higher, while there were no differences in functional divergence related to tree species (Figure 5B,C). For trunk assemblages, none of the diversity indices differed with tree species (results not shown).

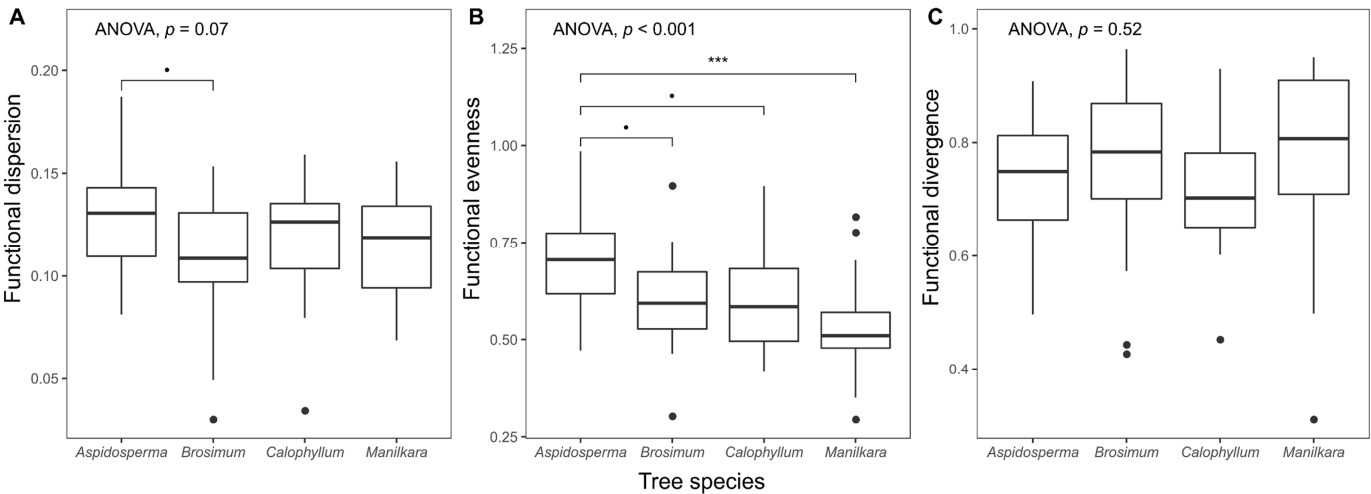

**Figure 5.** Functional diversity indices of crown epiphyte assemblages on focal host tree species. Boxplots show median, interquartile range and largest values within 1.5 times interquartile range below and above the 25th and 75th percentiles, respectively. Solid circles are outliers. Also given are *p*-values from analysis of variance, as well as the results of Tukey's post-hoc tests (Significance codes: "***" *p* < 0.001; "●": $0.05 \leq p \leq 0.1$).

## 4. Discussion

### 4.1. Trait Covariation

The bivariate relationships between leaf morphological and chemical traits of the epiphytes of this lowland forest were generally consistent with those found in non-epiphytic plants, e.g., in SLA, SRL, and leaf and root nutrient contents [32,50,52]. However, for other traits there were noteworthy deviations. While the positive relationship between $H_{max}$ and $LA_{max}$ was in accordance with the findings by Díaz et al. [52], the negatively correlation of $H_{max}$ with SLA was not: Díaz et al. [52] found no correlation between $H_{max}$ and SLA for herbaceous terrestrials. Similarly, a positive correlation of SLA and SD [80] could not be found in these vascular epiphytes. Díaz et al. [52] observed, expected from whole plant trait coordination [53], a negative correlation of SSD with SLA for herbaceous species, again at odds with our dataset. This discrepancy may be attributable to the fact that the requirements for stem morphology demanded by the main stem functions (water transport and mechanical support) are less uniform in epiphytes. Many epiphyte species have either very short and reduced stems (e.g., bromeliads, cespitose orchids) or a rhizome that grows appressed to the substrate. Moreover, for many epiphyte species (especially orchids with pseudobulbs) storage may be the dominant function. As a relevant trait in forestry, SSD data are widely available for trees but not for herbaceous species [20]. To include this trait in their global assessment, Díaz et al. [52] estimated SSD of herbs from LDMC. However, in our dataset LDMC and SSD were only weakly correlated (r = 0.24), indicating that for epiphytes such estimates would be unreliable.

The strong, negative correlation between SRL and RD is consistent with findings of many other studies (e.g., [51,55]) because with constant RTD, SRL scales inversely with root diameter squared [50] According to a postulated root economics spectrum, SRL should be negatively correlated with RTD. This expectation was met in a study with herbs [50] but not in analyses based exclusively or primarily on woody species [51,55]. In our dataset, the two variables were positively correlated. Similarly, the observed negative correlation between RTD and RD parallels previous findings of Weemstra et al. [55] and Ma et al. [51],

but not of Roumet et al. [50]. This suggests that in regard to root trait correlations vascular epiphytes resemble woody plants more than terrestrial herbs.

Given the large number of traits in the principal component analysis, it is remarkable that almost two thirds of the variation in the trait data was captured by only four dimensions. In the analysis of Díaz et al. [52] the first principal component was attributable to plant size and covarying traits while the second PC corresponded to the LES. In our PCA, these two dimensions also emerged as independent axes, but their relative importance was reversed (LES corresponded primarily to PC 1, while plant size was well represented by PC 3)—presumably because the range of plant sizes was much larger for their global dataset, including trees, than for our much smaller dataset on epiphytes that includes only herbaceous species of one forest. Root traits captured substantially more variation than plant size as indicated by their close correlation with PC 2. Kramer-Walter et al. [81] analyzed root, stem and leaf traits of tree seedlings via PCA. They found RTD to be highly correlated with the typical LES traits (SLA, LDMC), while SRL (and the negatively correlated RD) aligned along an orthogonal axis to these traits and was thus independent from RTD. Like their results, we found SLA, LDMC, and RTD and the strongly negatively correlated SRL and RD to be represented by different axes, but the axes were much less orthogonal.

The clustering of plant families in the ordination plots (Figure 3 and Figure S2C,D) suggests a phylogenetic signal in some studied traits. Particularly illustrative are Orchidaceae, which account for 44% of all taxa in our trait dataset—globally, they even account for c. 66% of all epiphyte species [82]. Orchids occupy a larger proportion of the ordination space than any other family but still clearly separate from most other families caused by their positioning along the RD/SRL axis due to their velamen radicum.

*4.2. The Relationship of Epiphyte Trait Patterns and Host Tree Species*

For the tree crown zone, our data yield strong evidence for trait matching between host tree and epiphyte species and, thus, species sorting exerted by species-specific environmental conditions on different tree species. Some of the observed CWM differences suggest that *Brosimum* has a more open crown than *Aspidosperma* and *Calophyllum*. Epiphytes on *Brosimum* with $C_3$ photosynthesis had higher $\delta^{13}C$ values (=higher intrinsic water use efficiency) and more succulent leaves (lower LDMC, higher $L_{th}$). Moreover, there was also a higher incidence of CAM plants on *Brosimum* as compared to *Manilkara*. Einzmann et al. [16] also found more CAM species on drought-deciduous vs. evergreen tree species. Although all four tree species in our study are evergreen, *Brosimum* has a relatively open crown and the highest temperature maxima of the studied trees [47]. In turn, light intensity was lowest on *Manilkara*, consistent with the low incidence of CAM plants on this tree species in the present study.

Microclimatic differences are only one aspect of trait matching, other factors are, e.g., bark properties and nutrient availability. For example, in addition to varying microclimate, differences in $\delta^{13}C$ and $L_{th}$ between epiphytes on *Brosimum*, *Aspidosperma*, and *Calophyllum* could also be related to differences in bark water holding capacity, which should increase with bark rugosity. The observed differences regarding bark rugosity are partly congruent with differences in CWMs. In accordance with the trait patterns, *Brosimum* has a very smooth bark that will dry out quickly after being wetted, while the bark of *Calophyllum* has deep fissures that will withhold moisture for longer, and *Manilkara* has by far the roughest bark [47]. Epiphytes on *Manilkara* had, on average, higher stomatal density, and smaller stomata than those on *Brosimum*.

We cannot explain the observed tree-related differences in the CWMs of root morphological traits, but they may reflect differences in water and nutrient availability. Within our epiphyte assemblages, high RD values indicate a velamen radicum (Figure 5C), presumed to be advantageous for effective water (and nutrient) uptake in tree canopies [65]. Therefore, we expected that epiphyte assemblages on tree species that pose stronger challenges regarding water supply (with low bark water holding capacity or open crown) would host a larger proportion of individuals with such large diameter roots. According to observed

patterns regarding epiphyte leaf traits on tree species and direct measurements of tree traits, we would thus expect the highest RD values on *Brosimum*. This expectation did not bear out: epiphytes in *Brosimum* had the lowest RD and highest SRL CWMs, while high RD were observed in *Aspidosperma*. Possibly, the hypothesized relationship between RD and canopy openness is simply not well justified—such a relationship was not found either for orchids along the vertical gradient in the forest [24].

Most CWMs of mass-based epiphyte leaf and root nutrient traits differed with tree species (exceptions were leaf K and Mg and root Ca). These traits may directly reflect the nutrient availability of the substrate, e.g., be determined by bark and leaf nutrient leaching. Unfortunately, we have no information on nutrient availability in the studied tree species to assess this notion, but tree species composition has been shown to strongly affect nutrient fluxes in throughfall and stemflow in tropical forests [83,84]. Foliar $\delta^{15}N$ is generally positively correlated with both, LNC and N availability [85], but we found no tree-related differences in assemblages regarding this trait, possibly indicating that the pattern regarding LNC is not caused by differences in N availability. Tissue nutrient concentrations are not merely a function of nutrient supply as they also covary with morphological traits of an organ. For example, mass-based leaf N and P were generally strongly correlated with SLA [32]. In the case of roots, there was a relatively tight correlation between RD and root K and Mg concentrations (r = 0.58 and 0.51, respectively). This could explain the high CWMs of RKC and RMgC on *Aspidosperma*, which hosted the assemblages with the highest RD.

The measurements of the SPAD chlorophyll meter correlate with chlorophyll concentration per area, (e.g., in [86]). Given that Petter et al. [3], using the same device, found only a weak (positive) relationship for LChl along the vertical gradient in the forest and LChl of sun and shade leaves are generally comparable [87], we expected at most a weak difference among tree species. In contrast, *Brosimum* had significantly lower CWMs of LChl than the other tree species. However, we note a bias because the SPAD device can be used only for leaves with <1.2 mm thickness and the most succulent leaves were excluded.

Besides trait averages of crown assemblages, functional diversity was also affected by tree species, albeit weakly. Paralleling the patterns found for CWMs, the strongest difference in functional diversity was observed between *Aspidosperma* (highest) and *Brosimum* (lowest). The trait spread of epiphyte assemblages on *Aspidosperma* was, on average, higher than on *Brosimum* (indicated by the functional dispersion index), where assemblages were less dominated by certain trait values than on the other tree species (indicated by functional evenness). This suggests trait convergence on *Brosimum* (more adverse conditions for epiphytes impose a stronger environmental filter) and a greater diversity of growing conditions or more beneficial conditions on *Aspidosperma*. The first notion is congruent with the finding that *Brosimum* has the lowest epiphyte accumulation rate of the four study species after accounting for tree growth rates [47]. However, in contrast to this conclusion, the previous study suggested also relatively adverse conditions on *Aspidosperma*, as this species generally had a lower epiphyte load compared to the other study species and ranked lower than *Calophyllum* and *Manilkara* in terms of epiphyte accumulation with time.

The expectation that host specificity depends on the vertical position on a tree was clearly supported. In contrast to crown assemblages, there were no differences in FD, and only one out of 28 traits showed differences regarding CWMs for trunk assemblages. This suggests that differences in crown architecture among tree species resulting in microclimatic differences in the upper part of the trees are more important for host specificity than differences in bark structure, which should be stronger in the lower part of a tree. Moreover, this finding supports the suggestion of Mendieta-Leiva and Zotz [48]: epiphyte host specificity should be analyzed separately for the different tree zones—at least in tall lowland rainforests with strong microclimatic differences between upper canopy and understory. However, it should be noted that the absence of significant differences of epiphyte CWMs among tree species on the tree trunk coincides with much lower numbers of epiphytes in this zone.

By calculating CWMs and FD indices based on epiphyte species trait averages, the present study considers only the part of the possible trait–environment relationships that is taxonomically constrained, as we calculate CWMs from average trait values per species, and thus ignore possible intraspecific adjustment of trait values along environmental gradients. For our research question (epiphyte host specificity), this approach is reasonable because specificity in species interactions can only result from differences at the species level. However, trait-environment relationships can also be caused by acclimation and sorting of genotypes causing intra-specific trait variation. For example, Einzmann et al. [16] found intraspecific differences for some combinations of the studied leaf traits and epiphyte species. When intra- and interspecific trait-environment relationships have the same direction [3], patterns as in the present study would only be reinforced by the inclusion of the intraspecific level. However, the assumption of an identical directionality does not always hold true as demonstrated by Kichenin et al. [88], who found several leaf traits (SLA, LNC and LPC) to respond with opposing directions at the inter- and intraspecific level along an elevational gradient. Therefore, future studies should also consider the intraspecific level.

Compared to previous trait studies with vascular epiphytes, the present study included an unprecedented large number of trait types. However, traits related to dispersal and seed adherence, which could be very important in the context of epiphyte host specificity, were not assessed. For example, diaspore size as well as structures that aid in the attachment to the bark may cause distributional biases among tree species with different bark types. It would certainly be worthwhile for future studies to assess the role of these traits.

## 5. Conclusions

Our study found intriguing indications for differences between epiphytes and terrestrial plants regarding trait covariation. However, even though our dataset comprises almost 100 epiphyte species, it is still small compared to the large datasets on which trait covariation analyses are often based. Future studies with larger datasets will show whether our results really identify general functional adaptations needed to thrive in the aerial habitat.

We found evidence for habitat-quality differences between host trees leading to trait filtering of epiphytes. In the tree crown, more than two thirds of the studied CWM traits of epiphyte assemblages differed significantly among the focal tree species. This finding is an important piece of the puzzle, as it gives us more certainty that epiphyte host specificity exists in high diversity systems such as the studied forest, although we were not able to fully explain the observed trait patterns with the available information regarding trait differences of the focal tree species. However, future studies with a stronger simultaneous focus on important tree traits should allow a full functional understanding of epiphyte host specificity.

**Supplementary Materials:** The following are available online at https://www.mdpi.com/1424-2818/13/2/97/s1, Figure S1: Correlation matrix for 26 traits, Figure S2: Principal component analysis of functional traits (excluding stomatal and root traits) of the epiphytic community at study site, Figure S3: CWMs of different traits on stems of focal tree species, Figure S4: CWMs of different traits in crowns of focal tree species. Table S1: Principal component analysis of functional traits, Table S2: Principal component analysis of CWMs functional traits, Table S3: Results of ANOVAs and pairwise comparisons of CWMs of traits on tree species for crown and trunk assemblages, File S1: References for trait-trait and trait-environment relationships depicted in Figures 1 and 2.

**Author Contributions:** Conceptualization, K.W. and G.Z.; Methodology, K.W.; Formal analysis, K.W.; Resources, K.W., G.Z., and W.W.; Data Curation, K.W.; Writing—original draft preparation, K.W.; Writing—review and editing, K.W., G.Z., and W.W.; Project administration, K.W.; Funding acquisition, K.W. All authors have read and agreed to the published version of the manuscript.

**Funding:** This work was funded by the Deutsche Forschungsgemeinschaft (WA 3936/1-1).

**Institutional Review Board Statement:** Not applicable.

**Informed Consent Statement:** Not applicable.

**Data Availability Statement:** Upon publication, data will be made publicly available in the TRY database.

**Acknowledgments:** This study would not have been possible without the dedicated help of many. Chrissi Göbel, Florian Masche, Paula Morales, Lilisbeth Rodríguez, Sebastian Schäfer, and Tizian Weichgrebe helped with carrying out the epiphyte census and sample processing in Panama. Samples were also processed and analysed in Oldenburg by Karoline Hots, Valerie-Manon Eppert, Niels Kappert, Katharina Lindefjeld, Norbert Wagner, and Malte Weber. Michael Kleyer (Oldenburg) kindly let us use his WinRhizo software for root analyses. We thank the Smithsonian Tropical Research Institute (STRI) for providing the necessary lab and field infrastructure and the Republic of Panama for making its natural resources available (research permits: SE/P-32-16 and SC/P-3-18).

**Conflicts of Interest:** The authors declare no conflict of interest. The funders had no role in the design of the study; in the collection, analyses, or interpretation of data; in the writing of the manuscript, or in the decision to publish the results.

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
