# Peer review of "Functional Traits of a Rainforest Vascular Epiphyte Community: Trait Covariation and Indications for Host Specificity"

_diversity, doi:10.3390/d13020097_

Round 1
Reviewer 1 Report
Dear Authors,
Your study of the functional traits of a rainforest vascular epiphyte community is very detailed and interesting. I definitely recommend that the manuscript be accepted, although I have a couple of really small suggestions and a few questions, which may contribute to a better discussion and perspectives for future research. It has been a pleasure to read your manuscript.
Abstract: Can you summarize the most important conclusion of your research and state it in one sentence?
Introduction:
In general, I think the introduction may be shorter.
Has the role of competition on the functional traits of a rainforest vascular epiphyte community been studied? Space competition is known to be very important in orchids. Perhaps you can write a section about it in the introduction, and suggest it in the discussion as well.
Materials and methods
Lines 234-238. Adjust the font size.
Can you define the type of forests in which you did the research, i.e. at the community level or the detailed type of habitat? In what period did you perform sampling? Why did you choose noted 4 tree species? What was the sample area and did you use the methodology already used in that sense or did you come to that area yourself?
Line 240: 99 species. Is this the result? If so, then replace this data to the results section.
Lines 310-315. I would not say that this is a text that corresponds to the methodology section.
Results
Lines 455-467. Refer to the appropriate table or figure somewhere.
Discussion
Start the discussion by citing some papers. In the first sentence, you write without referring to references to other works.
Can you compare your data with data related to terrestrial plants, especially orchids? I propose two papers studying the influence of ecological factors on the patterns of distribution, composition, and abundance of terrestrial orchids.
Djordjević, V., Tsiftsis, S., Lakušić, D., Jovanović, S., Jakovljević, K., Stevanović, V. (2020): Patterns of distribution, abundance and composition of forest terrestrial orchids. – Biodiversity and Conservation 29: 4111–4134. DOI: 10.1007/s10531-020-02067-6
Djordjević, V., Tsiftsis, S. (2020): The Role of Ecological Factors in Distribution and Abundance of Terrestrial Orchids. In: Mérillon, J.-M., Kodja, H. (eds.): Orchids Phytochemistry, Biology and Horticulture, Reference Series in Phytochemistry. Fundamentals and Applications. – Springer Nature Switzerland AG, 1-71. DOI: 10.1007/978-3-030-11257-8_4-1
When it comes to orchids, can you link your results to the CSR population strategy defined by Grime?
Based on your results, can any implications for conservation be derived? To reiterate, can competition have an important impact on the results obtained?
Author Response
Dear Authors,
Your study of the functional traits of a rainforest vascular epiphyte community is very detailed and interesting. I definitely recommend that the manuscript be accepted, although I have a couple of really small suggestions and a few questions, which may contribute to a better discussion and perspectives for future research. It has been a pleasure to read your manuscript.
Thank you very much for this very positive reception of our paper.
Abstract: Can you summarize the most important conclusion of your research and state it in one sentence?
We think that we did this already – if you want to pick one sentence in the abstract it would be “we found strong evidence for environmental filtering of epiphyte traits, but only in tree crowns”. See also our Graphical Abstract with the very same message.
Introduction:
In general, I think the introduction may be shorter.
Since this is the first major study on trait matching in vascular epiphytes we feel that the detailed introduction is warranted and would like to leave it as it is
Has the role of competition on the functional traits of a rainforest vascular epiphyte community been studied? Space competition is known to be very important in orchids. Perhaps you can write a section about it in the introduction, and suggest it in the discussion as well.
We now briefly mention that in our system competition is not likely to play an important role: “Competition, however, does apparently not play an important role as it does in terrestrial systems”
Materials and methods
Lines 234-238. Adjust the font size.
The problem does not appear in our Word document.
Can you define the type of forests in which you did the research, i.e. at the community level or the detailed type of habitat? In what period did you perform sampling? Why did you choose noted 4 tree species? What was the sample area and did you use the methodology already used in that sense or did you come to that area yourself?
We now cite a paper that describes the forest and one that describes the epiphyte community in that forest.
The remaining questions are now answered in the text:
- We chose four tree species, all of them tall growing trees (reaching ca. 35-40 m) and among the most abundant species at the study site, which allowed for a large replication.
- All sampled trees were located within a continuous area of ca. 40 ha.
- Sampling took place during the wet rainy season (July-November 2018).
Concerning your question about the reason why the sampling area was chosen as such: this area contained a minimum of ca. 20 tall growing individuals of all four tree species.
Line 240: 99 species. Is this the result? If so, then replace this data to the results section.
Although this is also a “result”, we need this number here because we describe our sampling scheme with different species numbers. Without the information that the total is 99 species, one would be lost in section 2.2. Therefore, we do not want to change anything here.
Lines 310-315. I would not say that this is a text that corresponds to the methodology section.
We disagree. We need this information to justify our methodological approach.
Results
Lines 455-467. Refer to the appropriate table or figure somewhere.
We actually referred to Figure S3 in line 447 already. All the rest also refers to this figure. We now mention Fig. S3 once more, but refrain from doing this even more often.
Discussion
Start the discussion by citing some papers. In the first sentence, you write without referring to references to other works.
We now cite some relevant other papers.
Can you compare your data with data related to terrestrial plants, especially orchids? I propose two papers studying the influence of ecological factors on the patterns of distribution, composition, and abundance of terrestrial orchids.
Djordjević, V., Tsiftsis, S., Lakušić, D., Jovanović, S., Jakovljević, K., Stevanović, V. (2020): Patterns of distribution, abundance and composition of forest terrestrial orchids. – Biodiversity and Conservation 29: 4111–4134. DOI: 10.1007/s10531-020-02067-6
Djordjević, V., Tsiftsis, S. (2020): The Role of Ecological Factors in Distribution and Abundance of Terrestrial Orchids. In: Mérillon, J.-M., Kodja, H. (eds.): Orchids Phytochemistry, Biology and Horticulture, Reference Series in Phytochemistry. Fundamentals and Applications. – Springer Nature Switzerland AG, 1-71. DOI: 10.1007/978-3-030-11257-8_4-1
Well, comparing our data with those of terrestrial plants is a major part of our discussion. Another issue a more fine-scaled analysis like a comparison of particular taxa like orchids. Thanks for pointing out these publications in that context. However, after reading them we are not really convinced that they fit in our paper. First, we are looking at epiphytes at large, second, the suggested papers are not about vegetative traits. We do agree that directly comparing epiphytic and terrestrial species of a given family would be potentially highly informative, but then there would be two potential options:
Compare them locally – we have no data for any local terrestrial member of orchids or any other family in our data set, so this does not work for us.
Compare regionally or even globally – for this we would have to compile all available data, which would be a major endeavor . Our paper is already rather long and adding this would be way too much.
Thus, we acknowledge this suggestion, but do not feel that we should do this in this paper.
When it comes to orchids, can you link your results to the CSR population strategy defined by Grime?
This is really a different story. Our paper is about trait matching between epiphytes and hosts and not about CSR. But this is an interesting which we will follow up in another paper that is currently in progress.
Based on your results, can any implications for conservation be derived? To reiterate, can competition have an important impact on the results obtained?
We already responded to the request for mentioning competition (see above). We do not mention conservation because this was outside the scope of our study and any reference to conservation would be far-fetched in our opinion.
Reviewer 2 Report
Dear authors,
I congratulate you on your very interesting and well-written manuscript. There are some minor points that need clarification (e.g., the number of traits you used in your analysis) and one suggestion: it would be really nice if you would be willing to check for the existence of phylogenetic signal in your data, as you hypothesize in line 634.

Author Response
Dear authors,
I congratulate you on your very interesting and well-written manuscript. There are some minor points that need clarification (e.g., the number of traits you used in your analysis) and one suggestion: it would be really nice if you would be willing to check for the existence of phylogenetic signal in your data, as you hypothesize in line 634.
Thanks for your positive comments. As far as the phylogenetic signal is concerned, please refer to the last comment below.
Please note: all line numbers refer to this pdf document
L 47 Such as? Please give at least an example here.
we now give 3 examples
L 91 Here you state that you used 28 traits, but in Table 1 there are 27 traits. Did I miss something here?
The legend clearly states that the 28th (categorical) trait is not listed. We added “28” now to the first sentence of the legend so that this misunderstanding will be avoided.
Table 1 Since you define these two abbreviations in the table, please consider stating here what they stand for.
We changed the sentence accordingly
Table 1 I may have missed something, but why there are some traits with a smaller number of observations compared to others?
That can be confusing (although later explained in Materials and Methods). We now explain it: “Sample size varies because it was not possible to determine all traits in every species.”
Figure 2: Please consider adding: "Abbreviations as in Table 1."
Now added as: “For a full list of traits and abbreviations see Table 1.”
L 217 Maybe move lines 90-100 here then? Because it seems as going back and forth regarding your aims. Or maybe I have missed something? Maybe slightly rephrase lines 90-100?
We think that these two paragraphs serve different goal. The first is at the beginning of the introduction and starts the detailed description of the framework of our study, while the second is the wrap-up of the goals. So we would like to keep both where they are.
L 231 Maybe instead of coordinates, consider providing a map presenting where the San Lorenzo forest reserve is.
We think that giving coordinates is common practice in a scientific paper. Anyone interested in a map can easily google either St Lorenzo or enter the coordinates.
L 235 How and why did you choose these four species? Is there a particular reason for it? Are they the most abundant trees? Are the ones with the higher epiphytic species richness? Could you please clarify this?
We chose these four species based on three criteria: they should have comparable final size, be very abundant to allow for sufficient replication and differ in their bark structure. We added this information in the text.
L 236 Font size is different here.
This is odd – in our word documents the font is not different from the rest of the text
L 257 Ah I see now why n varied in Table 1. But again you should probably state this.
Already changed
L 377 Since your data did not have a normal distribution, maybe the Spearman rank index would be more suitable.
In the preceding paragraph we mention that trait data did hardly deviate from normal or were log-transformed to achieve normality. Thus, it seems completely appropriate to use parametric statistics.
L 379 Space needed here
Space added
L 389 Did you try to impute these values? Johnson, T. F., Isaac, N. J., Paviolo, A., & González‐Suárez, M. (2021). Handling missing values in trait data. Global Ecology and Biogeography, 30(1), 51-62.
L 407 Am I missing something, but weren't these traits 28 (line 90)? In Table 1 though you mention 27 traits. Could you please clarify this?
After already responding to this issue and changing the legend of Table 1, this should no longer be an issue
Figure 4 Please clarify how many traits are included in the analyses, since they seem to hover between 27 and 28.
We realize that having different numbers of traits in the analyses can be confusing, but we really made an effort to be explicit how many traits were used and why. Here, e.g., we use 28 traits and state this, in Fig. 3 there are 26 and we also state this.
Figure 4 Please also consider adding: "Traits abbreviations as in Table 1."
Thanks for pointing this out. We added such a statement and did so as well in Figure 3.
L 605 Is there a reference for this?
We cited a relevant reference (Roumet et al. 2016).
Line 635 Yes, but did you test for something like this? It would be nice to run a simple phylogenetic analysis with e.g., the phylosignal R package and it won't take too much time to run it. Keck, F., Rimet, F., Bouchez, A., & Franc, A. (2016). phylosignal: an R package to measure, test, and explore the phylogenetic signal. Ecology and Evolution, 6(9), 2774-2780.
Thanks for this suggestion, but we came to the conclusion that we would like to refrain from such an analysis. We do not expect that phylogenetically independent contrasts will add much insight on top of what we know: For example, we already know that orchids and aroids have a velamen, which will show in root traits, we know that orchids and bromeliads have a large proportion of CAM species, we know that bromeliads have typically the lowers and aroids the highest leaf N-levels etc etc. The main focus of this study is on trait matching of epiphyte communities and host trees and we think that an additional analysis would not really add much to what we present. We therefore do not see an added value of removing phylogenetic signals from tree and epiphyte traits in a study of trait matching between host trees and epiphyte communities, which relies more on environmental drivers than on taxonomic signal, because accounting for phylogenetic signal would rather weaken any trait matching relations. We therefore did not apply such an analysis.
Reviewer 3 Report
This is well-done interesting study devoted to relationships between functional diversity of epiphytes and host specifity. The study was done in tropical forest in south America which is still unexplored thus results are very precious. Authors studied 99 species and took into account 28 traits. The paper is worthy publishinh I have only minor points to be addressed.
Figure 1 and 2. Authors wrote that ‘Numbers refer to publications that established these”, but in some cases two andmore numbers were given. It is unclear.
Line 107. Table 1. In table caption “For each of the 27 continuous traits “ So, how many traits were studied?
Line 257-258 “We determined 28 functional traits for 87-99 locally occurring 257 epiphyte species (Table 1).” This sentence is not clear. How many finally species were included?
Lines 377-391: Statistical analyses: different number of species were subjected to PCA analyses. What are reasons behinf that? No available data? Was it possible to include NAs i dataset?
Lines 430-436: Tukey’s tests could be performed only when ANOVA revealed significant results. No infomation about analysis of variance was provided and if data meet requirements for parametrical tests (normality of distribution).
Author Response
This is well-done interesting study devoted to relationships between functional diversity of epiphytes and host specifity. The study was done in tropical forest in south America which is still unexplored thus results are very precious. Authors studied 99 species and took into account 28 traits. The paper is worthy publishinh I have only minor points to be addressed.
Thanks for the positive evaluation
Figure 1 and 2. Authors wrote that ‘Numbers refer to publications that established these”, but in some cases two andmore numbers were given. It is unclear.
Sorry for the oversight. There was supposed to be a list of all these publications in the supplement. This list is now added as Appendix S1. With this list it should also be clear that there can be more than one publication backing up a given relationship.
Line 107. Table 1. In table caption “For each of the 27 continuous traits “ So, how many traits were studied?
This point has already been raised by reviewer 2 and has been dealt with (see above)
Line 257-258 “We determined 28 functional traits for 87-99 locally occurring epiphyte species (Table 1).” This sentence is not clear. How many finally species were included?
We agree that there is room for improvement! We now rephrased the sentence: “We determined 28 functional traits. Depending on the particular trait, data for 87 - 99 locally occurring epiphyte species could be collected (Table 1)”.
Lines 377-391: Statistical analyses: different number of species were subjected to PCA analyses. What are reasons behinf that? No available data? Was it possible to include NAs i dataset?
The large difference in number of species between the first and the second dataset results from Hymenophyllaceae (filmy ferns) not having roots and stomata. Thus, all these species are not present in the dataset with these traits. Including NA’s is not possible in PCA. Therefore, we decided to run a second analysis with more species (Hymenophyllaceae included) and fewer traits.
Lines 430-436: Tukey’s tests could be performed only when ANOVA revealed significant results. No infomation about analysis of variance was provided and if data meet requirements for parametrical tests (normality of distribution).
Thanks for pointing out that we did not explain this very well. To improve this, the text was modified accordingly. We performed an ANOVA or, in the case of heteroskedasticity, Welch’s heteroscedastic F test and reported the results alongside with the results of the post hoc tests (see Table S 3 and Figure 5). We assessed the approximate normal distribution of the residuals with Q-Q plots.